# Multifunctional Hybrid MoS_2_-PEGylated/Au Nanostructures with Potential Theranostic Applications in Biomedicine

**DOI:** 10.3390/nano12122053

**Published:** 2022-06-15

**Authors:** Thiago R. S. Malagrino, Anna P. Godoy, Juliano M. Barbosa, Abner G. T. Lima, Nei C. O. Sousa, Jairo J. Pedrotti, Pamela S. Garcia, Roberto M. Paniago, Lídia M. Andrade, Sergio H. Domingues, Wellington M. Silva, Hélio Ribeiro, Jaime Taha-Tijerina

**Affiliations:** 1Engineering School, Mackenzie Presbyterian University, Rua da Consolação 896, São Paulo 01302-907, SP, Brazil; thiago.malagrino@hotmail.com (T.R.S.M.); annapsgodoy@hotmail.com (A.P.G.); juliano.barbosa@mackenzie.br (J.M.B.); abn.guilherme@gmail.com (A.G.T.L.); nei.sousa@mackenzie.br (N.C.O.S.); jairojpedrotti@gmail.com (J.J.P.); pamela.s.garcia@gmail.com (P.S.G.); sergio.domingues@mackenzie.br (S.H.D.); helio.ribeiro1@mackenzie.br (H.R.); 2Departamento de Física, Universidade Federal de Minas Gerais, Avenida Presidente Antônio Carlos, 6.627, Belo Horizonte 31270-901, MG, Brazil; paniago@fisica.ufmg.br (R.M.P.); lidia.nanobmrg@gmail.com (L.M.A.); 3MackGraphe, Mackenzie Institute for Advanced Research in Graphene and Nanotechnologies, Rua da Consolação 896, São Paulo 01302-907, SP, Brazil; 4Departamento de Química, Universidade Federal de Minas Gerais, Avenida Presidente Antônio Carlos, 6.627, Belo Horizonte 31270-901, MG, Brazil; wellingtonmarcos@yahoo.com.br; 5Engineering Department, Universidad de Monterrey, Av. Ignacio Morones Prieto 4500 Pte., San Pedro Garza García 66238, NL, Mexico; 6Engineering Technology Department, University of Texas Rio Grande Valley, Brownsville, TX 78520, USA

**Keywords:** nanotechnology, hybrid nanostructures, molybdenum disulfide, PEGylated MoS_2_, gold nanoparticles, theranostic, biocompatible

## Abstract

In this work, flower-like molybdenum disulfide (MoS_2_) microspheres were produced with polyethylene glycol (PEG) to form MoS_2_-PEG. Likewise, gold nanoparticles (AuNPs) were added to form MoS_2_-PEG/Au to investigate its potential application as a theranostic nanomaterial. These nanomaterials were fully characterized by scanning electron microscopy (SEM), transmission electron microscopy (TEM), X-ray diffraction (XRD), photoelectron X-ray spectroscopy (XPS), Fourier-transformed infrared spectroscopy (FTIR), cyclic voltammetry and impedance spectroscopy. The produced hierarchical MoS_2_-PEG/Au microstructures showed an average diameter of 400 nm containing distributed gold nanoparticles, with great cellular viability on tumoral and non-tumoral cells. This aspect makes them with multifunctional characteristics with potential application for cancer diagnosis and therapy. Through the complete morphological and physicochemical characterization, it was possible to observe that both MoS_2_-PEG and MoS_2_-PEG/Au showed good chemical stability and demonstrated noninterference in the pattern of the cell nucleus, as well. Thus, our results suggest the possible application of these hybrid nanomaterials can be immensely explored for theranostic proposals in biomedicine.

## 1. Introduction

Over the last decades the application of nanomaterials in biomedicine has been raised, especially for biosensing and drug delivery. However, the main feature that must be considered to a given material be used in biomedical proposals is its physicochemical stability and biological safety [1]. In this context, molybdenum disulfide (MoS_2_) and its derivatives have attracted attention due to its excellent biocompatibility and simple synthesis [2]. The 2D nanomaterials have been used in the diagnosis and therapy of cancer [3] especially when they are combined with gold nanoparticles [4,5,6,7]. For instance, graphene and other nanocarbons [3,8,9], hexagonal boron nitride (h-BN) [1,10], black phosphorus (BP) [11], tungsten sulfide (WS_2_) [12], molybdenum disulfide (MoS_2_) [13,14,15], among others, have gained prominence mainly due to their huge biological applications [13]. Molybdenum disulfide is an important material because, in addition to being widely studied for use in cancer treatment [16,17], such as the photodynamic (PDT) and the photothermal (PTT) therapies, it is an abundant material with appreciated mechanical, thermal and electrical properties, that makes it also economically accessible [18,19]. Although MoS_2_ is considered naturally biocompatible [20], it is common to seek an improvement in its colloidal stability for in vivo studies to achieve an increased penetration and dispersibility in the tissues of interest [13,21]. In some cases, the biochemical stability is reached by the modification of its surface, for instance with polyethylene glycol (PEG) as well as other chemical groups [5,6,13,16,22,23]. This functionalization process can allow its selectivity and effectiveness treatment in cancer cells promoting, for example localized and controlled hyperthermia by near-infrared radiation (NIR) [24,25]. It has the potential to replace conventional treatments such as chemotherapy [17]. Due to the ability of MoS_2_ to convert light-to-heat its photothermal properties have been studied for cancer treatment, alone or in combination with different therapies [26]. For instance, an in vivo study that used human breast cancer xenograft model in nude mice, treated with PEGylated nanocubes containing Fe_3_O_4_@MoS_2_ and Doxorubicin for MRI-guided chemo-photothermal therapy showed decreased tumor volume after 24 h of injection [27]. Shi et al. observed in MCF-7, a human breast cancer cell line, an increased apoptosis death due to the ability of MoS_2_-PEG functionalized with Doxorubicin to evade endosomes via NIR light irradiation [28]. In the same way, AuNPs also are the most used element for nanoparticles-based biomedicine applications. One example is its fluorescence signals enhancement used for diagnostic purpose [29]. To our knowledge, there are few studies that report the use of MoS_2_-PEG/Au for theranostic applications [30], reported in the literature, despite the significant advances in nanomaterials science applied in cancer diagnosis and therapy [5,13,17,23,26,31,32]. The use of hybrid nanostructures still lacks intense exploration in this specific area. In this work, hierarchical flower-like MoS_2_-PEG or MoS_2_-PEG/Au were produced and fully characterized. Moreover, the impact of these nanomaterials in the cellular viability was investigated in vitro and data show non-toxicity associated with both, suggesting their potential application in biomedicine.

## 2. Materials and Methods

### 2.1. Materials

#### 2.1.1. Reagents

The reagents, ammonium molybdate, (NH_4_)_6_Mo_7_O_24_ (99.98%), Polyethylene glycol 20,000, CH_3_O(CH_2_CH_2_O)nCH_3_ (PEG-2000), Thioacetamide, C_2_H_5_NS (98%), Ethanol, C_2_H_5_OH (99.98%), acetone, C_3_H_6_O (99.98%), Sodium borohydride, NaBH_4_ (98%), Hexadecyltrimethylammonium bromide, C_19_H_42_BrN (CTAB, 99%), Silver nitrate, AgNO_3_ (99%), Gold (III) chloride trihydrate, HAuCl_4_·3H_2_O (99.99%), L-ascorbic acid and C_6_H_8_O_6_ (99%), were obtained from Sigma Aldrich, Brazil.

All aqueous solutions were prepared with ultrapure water (resistivity > 18.2 MΩ.cm) generated by a Millipore Milli-Q system.

#### 2.1.2. Cell Lines and Reagents

All cell lines used (human squamous cell carcinoma A431, human pharynx carci-noma FaDU and green monkey kidney Vero CCL-81) were obtained from ATCC (Manas-sas, VA, USA) and grown at 37 °C with 90% humidity and 5% CO_2_ incubator in DMEM High Glucose containing 1 mM sodium pyruvate from Gibco (Grand Island, NY, USA); they were also supplemented with 5% Fetal bovine serum from LGC Biotechnology (Cotia, SP, Brazil).

Trypsin solution from Gibco (New York, NY, USA). Hoechst 33,342 was obtained from Sigma Aldrich (New York, NY, USA). T25 Tissue culture flasks and multi-well plates were obtained from Kasvi (Curitiba, PR, Brazil) and TPP (Trasadingen, Switzerland). Resazurin viability kit was purchased from Sigma (St. Louis, MO, USA).

### 2.2. Methods

#### 2.2.1. MoS_2_-PEG Synthesis

The material was synthesized with hydrothermal method. In a typical experiment, ammonium molybdate (1.80 g) and PEG-2000 (1.0 g) were dissolved in 60 mL deionized water under stirring. Thioacetamide (1.80 g) was added into the solution and the mixture was stirred for 1 h. Then the mixture was transferred into a 100 mL autoclave and reacted for 3 h at 180 °C in an oven. After the autoclave cooling, the precipitate was collected by filtration and washed several times with Milli-Q water, ethanol and acetone. The material was dried in an oven at 100 °C for 4 h and MoS_2_ microspheres were obtained.

#### 2.2.2. Au Nanoparticles Production

The production of the gold nanoparticles (AuNPs) followed the method described by Meireles et al. [33]. The stock solution was prepared by mixing 5 mL of CTAB (0.2 M) and 5 mL of HAuCl_4_·3H_2_O (0.0005 M). Then, 60 μL of ice-cold NaBH_4_ (0.010 M) was added to the mixture, resulting in a brownish-yellow solution. Vigorous stirring was kept for 2 min at room temperature. The growing solution was prepared by adding CTAB (0.20 M), AgNO_3_ (0.02 M) and HAuCl_4_·3H_2_O (0.005 M). Then, 210 μL of L-ascorbic acid (C_6_H_8_O_6_) (0.4 M) was added to the system as a moderate reducing agent changing the color of solution from brownish yellow to colorless. Finally, 300 μL of stock solution was added to the growing solution, kept stirring for 20 min, and gradually changing to a deep red color.

#### 2.2.3. MoS_2_-PEG/Au Synthesis

The MoS_2_ (500 mg) sample was dispersed in 70 mL of a mixture containing 350 mg of gold nanorods for 30 min in a bath ultrasound operating at power of 20 W. Then, the mixture was transferred for a polytetrafluoroethylene (PTFE) vessel. The reaction vessel was attached to an autoclave and the incorporation was carried out in an oven at a temperature of 180 °C for two hours.

#### 2.2.4. Sample Preparation for Electrical Measurements

For sheet resistance measurements and contact angle measurements MoS_2_-PEG and MoS_2_-PEG/Au samples were prepared by mechanical compaction using a Shimadzu, SSP-10 A, P/N 200–64,175 hydraulic. The materials were pressured at 50 kN for 2 min, which resulted in disks of 13 mm in diameter and 2 mm in thickness, which can be seen in Appendix A. For electrochemical measurements flexible graphite sheets, 1 mm thick, were cut into 2 × 4 cm^2^ rectangles which served as conductive substrate for the electrodes. The electrodes were prepared for use through a procedure that included cleaning the graphite surface with the aid of a soft paper and washing with 92% ethanol solution (*v*/*v*), followed by drying with N_2_ flow. Then, around 1 cm^2^ of the graphite electrodes were modified by drop-casting 1 mL dispersions (1.0 mg mL^−1^) of MoS_2_-PEG and MoS_2_-PEG/Au in isopropyl alcohol. The samples were dried in an oven at 150 °C for 24 h.

#### 2.2.5. Cellular Viability Based on Resazurin Reduction Assay

Cellular viability was assessed by performing viability method according to the manufacturer’s instructions (Sigma, St. Louis, MO, USA). Briefly, the cells were seeded at 1 × 10^4^ cells per well in 96-well flat-bottom plates (Kasvi, Curitiba, PR, Brazil) and incubated for 24 h at 37 °C with 5% CO_2_ atmosphere. Cells were exposed to four different concentrations of MoS_2_-PEG and MoS_2_-PEG/Au, respectively, 100 µg/mL, 50 µg/mL, 10 µg/mL and 5 µg/mL. For death control Triton 100 × 10% was used. Each sample was dispersed in ultrasonic bath for 60 min and left exposed to each cell line for 24 h and 48 h. Subsequently, 10 µL per well of the reagent resazurin were added to the cells already cultured (1:10 final dilution). The cells were kept under biochemical reaction for 4 h at 37 °C in the dark and the absorbance at 580 nm was measured in a spectrophotometer (Multiskan GO, Thermo Fisher Scientific, Waltham, MA, USA). All tests were performed at least two times. The fraction of viable cells in treated groups was calculated as a percentage of the untreated control group that was considered 100% viable.

#### 2.2.6. Hoechst Nuclear Staining Assay

To evaluate cell death patterns after nanomaterials treatment, a fresh staining solution containing Hoechst 33,342 (5 µg/mL) was used. Briefly, 1 × 10^4^ FaDU cells were seeded in a 96 well plate and after monolayer cell formation, cells were treated with 100 µg/mL of MoS_2_-PEG/Au for 24 h and 48 h, respectively. Triton 100× was used as death control. Before staining, growth medium was removed and 50 µL of staining solution were added to each well. After 60 min at 37 °C, CO_2_ incubator staining solution was removed and cells were fixed by formalin 10%. Cells were analyzed in an Evos FL cell image system microscope (Thermo Fischer, Waltham, MA, USA), (Belo Horizonte, Brazil). The stained cellular nuclei were counted by using ImageJ2 software (NIH, USA).

#### 2.2.7. Characterization

FTIR measurements were made with Thermo Nicolet 6700 equipment (São Paulo, Brazil). The spectra were collected in the ATR mode from 64 accumulations in the transmission mode and were adjusted, taking baseline corrections into account. Micro-Raman measurements were carried out using a Horiba Jobin Yvon iHR 550 spectrometer (São Paulo, Brazil). The 532.0 nm line of an Ar-Kr laser was used as the excitation source. UV–visible spectroscopy measurement of gold nanoparticles was carried out in a UV-2550 spectrometer (Shimadzu, Japan) (São Paulo, Brazil). The spectrum was collected from 1000 to 200 nm in water suspension, where more details can be seen in Appendix A. SEM characterization was made in a Carl Zeiss Field Emission Scanning Electron Microscope, model sigma VP equipped with an energy-dispersive X-ray (EDS) (São Paulo, Brazil). EDS spectra were obtained using a Bunker detector 410-M and software Quantax Espirit 1.9 (São Paulo, Brazil). The Peakfit v4 program was used to adjust the EDS spectra. The STEM images were carried out on a FEI TECNAI G2 (São Paulo, Brazil), microscope with a tungsten filament electron gun of 200 kV. XPS spectra were obtained using monochromatic Al Kα radiation (1486.6 eV). XPS spectra were acquired at room temperature on a VG Scientific Escalab-220-ixL (Belo Horizonte, Brazil). The detailed states of the different elements (C1s, O1s, Mo3p, Mo3d and S2p and Au4f) were obtained from fitting the XPS peaks by using Lorentzian and Gaussian functions. The sheet resistance and electrical conductivity of the nanostructures were measured in quintuplicate at room temperature by using an Ossila T2001A2 Four-Point Probe System (Sheffield, UK) (São Paulo, Brazil), with a probe space of 1.27 mm. Each material was evaluated by the smallest current signal that showed any response as voltage output. The applied current was 5 μA for both samples. The electrochemical measurements were carried out with a potentiostat Metrohm, model PGSTAT 128N (São Paulo, Brazil), controlled by software NOVA version 1.1.1, combined with a lab-made cell with three-electrode configuration, shown in Appendix A.

## 3. Results and Discussion

### Characterization of Flower-Like MoS_2_-PEG/Au

The morphology and microstructure of the as-synthesized MoS_2_ with polyethylene glycol (PEG) were characterized by SEM and STEM microscopy. SEM image (Figure 1a) shows that the structures are formed by uniform hierarchical microspheres of MoS_2_ with average diameter of about 400 nm. According to the literature [34], PEG can provide template structure in the synthesis, obtaining flower-like MoS_2_ microspheres. STEM images (Figure 2b,c) reveal that each microsphere is composed of several curled and ultra-thin nanosheets distributed on its surface, and the MoS_2_ interlayer spacing has approximately 0.54 nm.

Comparing the morphologies of the as-synthesized MoS_2-_PEG (Figure 2a–c) and MoS_2_PEG/Au by SEM (Figure 2b–d), it is noticeable that both structures appear as branched structure of aggregates with rough surface. However, when Au nanorods were added, microspheres exhibited less uniform shape showing a small difference in morphology in relation to the MoS_2_-PEG (see Figure 2d).

Figure 3a–c shows EDS mapping were made to evaluate the element distribution of the MoS_2_-PEG/Au sample. It can be observed that the elements Mo, S and Au are uniformly distributed in the studied area. The EDS spectrum (Figure 3d,e) reveals the dominance of Mo and the primary component of MoS_2_, which is sulfur. The presence of C comes from the PEG used as a “covering agent” of the nanoflake during the synthesis [35]. The peaks of Si and Al, which can be considered a type of impurity, are probably associated to the elements present in the sample holder. After modification with AuNPs (Figure 3e) a new peak referring to Au energy was identified [36]. Detailed discussion of the TEM and UV-vis of AuNPs can be seen in Appendix A.

The FTIR spectrum (Figure 4a) showed the main bands related to the presence of MoS_2_: 826.4 and 931.5 cm^−1^ [16,37]. The band at 826.4 cm^−1^ represents the stretching mode band of the Mo=O bond. Previous studies reported that when in contact with air, the surface of MoS_2_ can be partially oxidized and contains adsorbed water, Mo=O groups, sulfate and thiosulfate anions, among others [16,37,38,39]. At 931.5 cm^−1^, the band originating from the vibrational mode is characteristic of S-S bond [35]. The contribution of PEG begins to appear at 1083.5 cm^−1^, where the intense and sharp band points out to the C–O stretching mode, and at 1427.9 cm^−1^, with less intense but even strong band from the bending mode vibration of C–H. At, 2880.8 cm^−1^, a discrete band is showed from the weak stretching mode vibration of C–H. The results confirm that PEG acts as a wrapper on MoS_2_ nanoparticles [16,40,41,42]. As well the weak bands at 3021. 4 and 3179.1 cm^−1^ as the intense bands at 3616.5 and 3715 cm^−1^ correspond to the stretching mode of the inter and intramolecular hydrogen bonds and of the hydroxyl adsorbed from the atmosphere [40,41,42,43]. No significant changes were observed in the MoS_2_-PEG/Au spectrum related to MoS_2_-PEG, so it was suppressed in this work. For the MoS_2_-PEG nanomaterial, Raman spectroscopy (Figure 4b) generated the characteristic bands of MoS_2_ (at 371.5 and 462.8 cm^−1^) assigned to the E_2g_ and A_1g_ vibrational modes, respectively. Such modes, in turn, correspond to the E_2g_ (in-plane) vibrations of sulfur and molybdenum atoms, and to the A_1g_ (out-of-plane) vibrations of them [41]. In the regions of low frequency in Raman (from 329.4 cm^−1^), bands connected to vibrational modes of flexion between C–O and C–O–C were found, as well as low or medium PEG backbone vibrations. The band at 659.3 cm^−1^ is also derived from the C–C vibration of the PEG skeleton, as well as the intense band at 817.2 cm^−1^ [42,44,45]. The band at 992.7 cm^−1^ refers to the stretching and bending mode of the O–H bonds and the stretching of the C-C bonds, which is well defined and more intense for primary alcohols, as is the case with PEG [34,46,47,48,49]. The MoS_2_-PEG/Au hybrid presented the same bands as the MoS_2_-PEG, but with the additional band at 1493.0 cm^−1^ ascribed to γ(C–C) from PEG which was intensified due the presence of gold nanostructures [50].

The surface chemical characterization of the produced nanomaterials was performed by XPS. As shown in the survey spectra, Figure 5a, both MoS_2_-PEG and MoS_2_-PEG/Au samples exhibit the elemental Mo, S components as well as carbon and oxygen peaks. The Au4f peak was under the detection limit for MoS_2_-PEG/Au, suggesting that Au has been covered by the other elements. Figure 5b shows the C1s peak, indicating that the amount of carbon has increased five times after MoS_2_-PEG/Au synthesis, especially related to the C–O–C, C–OH and C–C species. The O1s peak for MoS_2_-PEG, see Figure 5c, exhibits a peak at 532.6 eV attributed to O–C, perfectly in line with the presence of PEG [51]. MoS_2_-PEG/Au shows, besides the O–C bond, a feature at 531 eV which can be attribute to a metal oxide (e.g., MoO_3_), for that sample. By examination of the Mo3d and S2p doublets of MoS_2_-PEG, the bottom spectra of Figure 5d,e, respectively, where it is possible to conclude that the MoS_2_ synthesis was successful. Peaks are very narrow as expected for a crystalline sample and the main peaks, Mo^4+^ 3d_5/2_ at 229.5 eV and S^2−^ 2p_3/2_ at 162.4 eV agree well with the binding energies found by other authors [52]. Besides that, only a very low contribution of Mo^6+^ attributed to molybdenum oxide and a broad (and low) S2p feature at ~169 eV associated with sulfate (S^6+^) species were observed. The synthesis with AuNPs, which involved bath ultrasound and thermal treatment in air, has importantly modified the surface near region of the sample. Now the Mo 3d_3/2_, 3d_5/2_ and S 2p_1/2_, 2p_3/2_ doublets are highly attenuated at the MoS_2_ binding energy positions, the Mo3d peaks were shifted to higher binding energies, where it can conclude that the surface has been oxidized. Two Mo3d_5/2_ peaks with binding energies of 232.6 eV and 233.6 eV are assigned to molybdenum oxide (MoO_3_) and molybdate species, respectively. The same occurs to S2p, it shrinks for MoS_2,_ and a sulfur doublet shows up at higher binding energy. The S2p_3/2_ line appearing at 169.6 eV is attributed to S^6+^ in sulfates (SO_4_)^2^, which can be explained by either MoO_2_(SO_4_) or Mo(SO_4_)_3_ formation at the surface of MoS_2_. Moreover, it was observed some sulfur loss at the surface near region, which may be due to the formation of volatile sulfur compounds during heat treatment.

Before the capacitance measurements, the electrical characteristics for MoS_2_-PEG and MoS_2_-PEG/Au hybrid microspheres were evaluated by sheet resistance, resistivity and conductivity measurements using the four-point probe technique. As shown in Table 1, the two composites exhibit low electrical conductivity. However, as expected, the incorporation of AuNPs in MoS_2_-PEG significantly decreased the resistivity of the hybrid system. The best electrical conductivity of the MoS_2_-PEG/Au nanostructures can be associated with junctions between Au nanoparticles in which the electrons are transferred more easily [53]. Additionally, the sulfur atoms over the outer layer of the MoS_2_ structure are suitable for the formation of strong Au-S bonds, which improves the charge transfer between sulfur and AuNPs [54,55].

Cyclic voltammetry (CV) measurements were performed to evaluate the charge storage capacity of both materials. Figure 6a–d shows the cyclic voltammograms obtained with (a) MoS_2_-PEG and (c) MoS_2_-PEG/Au electrodes (geometric area = 3.1 mm^2^) in 1.0 M Na_2_SO_4_ electrolyte solution over a potential range of −0.2 to 0.4 V at different scan rates. The current signal at the CV curves measured with both electrode materials increases with the corresponding scan rate and the voltammograms show a nearly rectangular aspect, suggesting the charging current is the dominant signal due to reversible adsorption of electrolyte ions on the surface of electrodes. The pattern of specific capacitance decreases for both electrodes (Figure 6b–d) as the scanning rate increases, which can be attributed to lower interactions between electrolyte ions and active sites of the electrode at higher scan rates [56] and as expect the best capacitive results was achieved at 1 mV s^−1^, being 181 F g^−1^ and 96 F g^−1^ for MoS_2_-PEG and MoS_2_-PEG/Au, respectively.

The low capacitance of MoS_2_-PEG/Au compared to MoS_2_-PEG was an unexpected fact, however when observing the values of the contact angle measurements of MoS_2_-PEG and MoS_2_-PEG/Au electrodes, Table 2, it is verified that the coating MoS_2_-PEG on the graphite sheet surface electrode presents greater wettability than MoS_2_-PEG/Au electrode. This fact may be related to the low electroactive area of the MoS_2_-PEG/Au electrode due to its lower wettability, not allowing the effective access of the electrolyte in the specific electroactive area [57].

The electrochemical impedance spectroscopy (EIS) technique is an important tool for evaluating the behavior of electrode material at the interface electrolyte-electrode [58]. To examine the performance with both electrode materials, the impedance was measured at 1 MHz to 0.040 Hz at +0.2 V vs. Ag/AgCl with an AC amplitude of 10 mV in a solution containing both [Fe(CN)_6_]^3−/4−^ redox couple 2 mM in KCl 0.2 M. The Nyquist plot is a plot of the real part of impedance (Z′) vs. imaginary part of impedance (−Z′′). Figure 7 shows the Nyquist plots for both electrodes. At the highest frequency range, the value at the real axis corresponds to the equivalent series resistance (ESR) and represents the contribution of the electrolyte resistance and the intrinsic resistance of the electrode material. As both electrodes are in the same electrolyte, the ERS reflects the difference in the electrode material.

As a comparison, the MoS_2_-PEG electrode shows an ERS value of 256.4 Ω while MoS_2_-PEG/Au presents a higher value 339.2 Ω, which can justify the better capacitive performance of the MoS_2_-PEG electrode. The charge transfer resistance, which is measured from the semi-circle along the real axis from high to medium frequency is also lower in the MoS_2_-PEG electrode than in the MoS_2_-PEG/Au one. At low frequency, the Nyquist plots showed a linear range with a slope of near 45°, known as Warburg resistance, where the frequency signal depends on the ion transportation in the electrolyte [57]. The examination of the curves at low frequency shows a smaller Warburg region for the MoS_2_-PEG electrode, suggesting a lower ion diffusion resistance in the structure of the electrode [59] while impedance performance of the MoS_2_-PEG/Au electrode shift along the *x*-axis toward more resistive values. Therefore, the better charge transmission and the ion diffusion through the electrode material structure help to explain the higher capacitance of the MoS_2_-PEG electrode.

The potential cytotoxicity of a given nanomaterial is a prerequisite for biomedical applications. In this study we assessed whether MoS_2_-PEG or MoS_2_-PEG/Au could be able to promote reduction of cellular viability in normal and tumoral cells under four different concentrations. The Green Monkey Kidney cell line VeroCCL-81 was used as non-tumoral model and the results applying MoS_2_-PEG showed discrete cellular viability reduction after 24 h when exposed to 10 µg/mL (2.1 ± 0.2)% and 5 µg/mL (5.2 ± 0.1)%, respectively, as shown in Figure 8a. On the other hand, cells exposed to MoS_2_PEG/Au demonstrated decreased cellular viability in all concentrations tested, such as (12.2 ± 0.3)% for 100 µg/mL, (19.9 ± 0.9)% for 50 µg/mL, (14.4 ± 0.4)% for 10 µg/mL and (2.5 ± 0.1)% for 5 µg/mL, respectively, compared to 65.5% ± 0.1% of the death control cells. After 48 h no cellular viability reduction was observed for cells exposed to MoS_2_-PEG and smaller decrease of cellular viability was observed in cells exposed to MoS_2_-PEG/Au at 100 µg/mL (3.2 ± 0.1)%, 50 µg/mL (5.5 ± 0.2)%, 10 µg/mL (5.0 ± 0.1)% and 5 µg/mL (6.9 ± 0.1)%, respectively, as shown in Figure 8c,d, suggesting a recovery of the cells in a time-dependent manner. Even though good dispersion in water small aggregations were observed over the cells as highlighted in Appendix A.

To study the effects of MoS_2_-PEG and MoS_2_-PEG/Au in the cellular viability of tumoral cells, the human squamous carcinoma A431 cell line and the human pharynx carcinoma FaDU cell line were used. For A431 cells no difference in the percentage of cellular viability was observed for MoS_2_-PEG in 24 h, as shown in Figure 9a. For 48 h there were a small decrease of the cellular viability with 100 µg/mL (5.2 ± 0.2)% and 50 µg/mL (5.8 ± 0.3)%, respectively, as observed in panel (b). A similar result was found when these cells were exposed to MoS_2_-PEG/Au at 48 h showing (3.2 ± 0.1)% decreasing for 100 µg/mL and (4.2 ± 0.2)% for 50 µg/mL, respectively, as shown in panels (c) and (d). No alterations in the cellular morphology were observed as shown in Appendix A.

Another tumoral model was used and MoS_2_-PEG showed a decrease of 3.8 ± 0.3% for 100 µg/mL and (11.8 ± 0.2)% for 50 µg/mL at 24 h, as shown in Figure 10a. At 48 h there was only (2.4 ± 0.1)% for 50 µg/mL, suggesting a recovery of these cells in a time-dependent manner and a decrease of (7.3 ± 0.2)% was noticed for 5 µg/mL as shown in panel (b). For MoS_2_-PEG/Au the most pronounced cellular viability decreasing were observed for 100 µg/mL (6.5 ± 0.9)% and 50 µg/mL (11.61 ± 0.12)% at 48 h, as shown in panels (c) and (d). After contact with both nanomaterials FaDU cells demonstrated morphological changing when compared to control cells, as shown in Appendix A. It is possible that these cells may present, somehow, plasma membrane reaction when in contact with charged materials, by promoting shape alterations. Then, due to the differences observed in their shapes and the decreasing of cellular viability the effect of 100 µg/mL of MoS_2_-PEG/Au in the number and shape of their nuclei were assessed by Hoechst 33,342 nuclear staining. Interestingly, no differences were observed neither in nuclei number nor in nuclei morphology even at the highest concentration applied, when compared to control cells as shown in Figure 11.

Our results demonstrated that both nanomaterials are biocompatible in normal cells, as well as in tumoral cells. Despite the high concentrations applied, for instance, 100 µg/mL and 50 µg/mL the cellular viability was kept above 80%. Moreover, at 48 h of contact with MoS_2_-PEG and MoS_2_-PEG/Au all cell lines presented very low damage what suggests the safety of these nanomaterials in vitro. Gold nanoparticles have been largely studied in biomedicine due to their well-known biocompatibility [60]. It seems that FaDU cell line changes its shape in contact with both MoS_2_-PEG and MoS_2_-PEG/Au. Nonetheless, this change in morphology did not alter cellular viability or nuclear number as demonstrated in Figure 11, corroborating the safety of MoS_2_-PEG/Au. Although both nanomaterials exhibited good dispersion in water they agglomerated in culture medium, maybe due to the presence of protein and salt. However, MoS_2_-PEG and MoS_2_-PEG/Au structures were able to adsorb in the cellular membranes and it is possible that a small amount of these nanomaterials carried into the cells which explains the alterations in cellular viability percentages. It was already shown that 2D MoS_2_ nanoparticles do not promote deleterious effects in cellular viability or induce genetic defects in HEK293f cells [20]. In this way, our data suggest the same behavior in other cellular models. Since neither MoS_2_-PEG and MoS_2_-PEG/Au themselves decrease cellular viability, they can be functionalized for theranostic purposes. We believe that MoS_2_-PEG/Au may possess potential application as a platform for theranostic, especially for cancer screening and photothermal therapy, because transition-metal dichalcogenides also enhance fluorescence through FRET and MoS_2_ nanosheets coated with folic acid had demonstrated efficient detection of miRNA-21 expression in a unique MCF-7 and HeLa cancer cells lines, working as a fluorescent nanoprobe. Based on the lack of cellular injuries demonstrated by our data both nanocomposites could have their surfaces modified to improve their dispersibility in aqueous solution. As an example, MoS_2_-PEG could be investigated as a new contrast agent for radiological imaging, and MoS_2_-PEG/Au could be functionalized to promote drug-delivery. The carefully physicochemical and biological characterizations of nanomaterials for biomedical applications are the first steps for a such proposal [61], and our work provides data enough to indicate MoS_2_-PEG/Au for these purposes. Then, this initial study paves the way for other works in nanobiomedicine, for instance by incorporating dyes and/or biomolecules to MoS_2_-PEG/Au by taking advantages of the excellent optical properties of AuNPs and MoS_2_ as well.

## 4. Conclusions

MoS_2_-PEG and MoS_2_-PEG/Au were successfully synthesized and widely characterized with different physicochemical techniques. By using TEM and SEM, it was possible to verify hierarchical flower-like structures, with a lateral size of up to 400 nm. Through the EDS and XPS studies, it was also observed that AuNPs were distributed throughout the material. MoS_2_-PEG/Au sample showed a lower capacitance value compared to MoS_2_-PEG; this fact may be associated with the low wettability of this electrode as observed in its contact angle. In this case, the presence of AuNPs could be related to increase the MoS_2_-PEG hydrophobicity, allowing only partial contact between the electrode and the electrolyte. However, in both cases the produced samples showed chemical stability and biocompatibility, due to the cell viability achieved above 80% for different times. Thus, its potential use as multifunctional nanomaterial for theranostic applications is an important subject that needs to be better explored.

## Figures and Tables

**Figure 1 nanomaterials-12-02053-f001:**
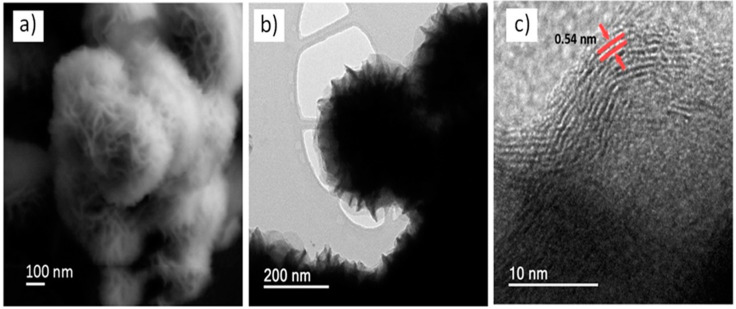
(**a**) SEM and (**b**,**c**) STEM images of the flower-like MoS_2_ microspheres using PEG in the synthesis.

**Figure 2 nanomaterials-12-02053-f002:**
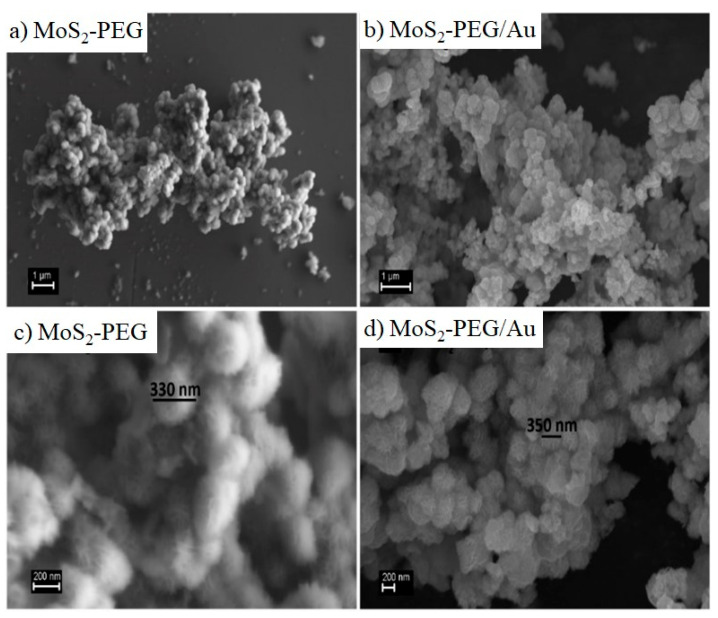
SEM images of the flower-like MoS_2_ microspheres using: (**a**,**c**) PEG and (**b**,**d**) PEG/Au from their synthesis process.

**Figure 3 nanomaterials-12-02053-f003:**
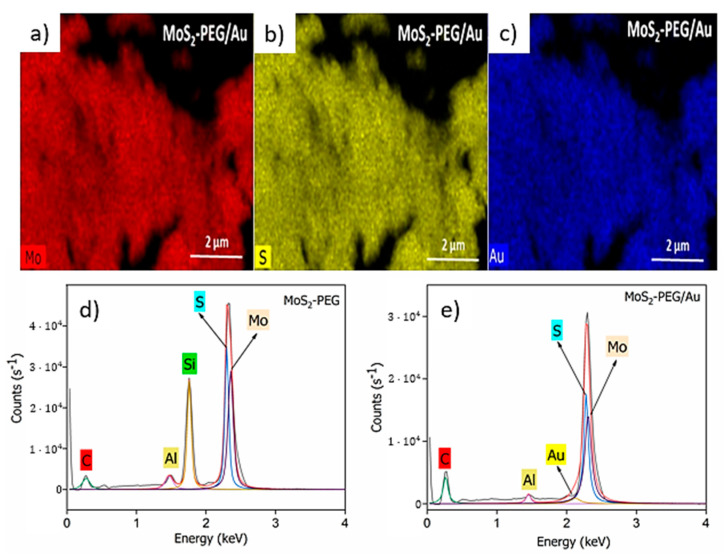
(**a**–**c**) Corresponding MoS_2_-PEG/Au EDS mapping with Mo, S and Au elements. EDS spectrum of (**d**) MoS_2_-PEG and (**e**) MoS_2_-PEG/Au.

**Figure 4 nanomaterials-12-02053-f004:**
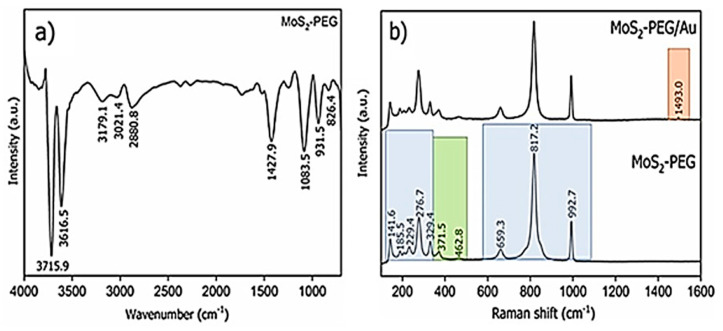
FTIR spectrum (**a**) of MoS_2_-PEG and Raman spectra (**b**) of MoS_2_-PEG (in black) and MoS_2_-PEG/Au. Characteristic bands regions referring to MoS_2_ (in green), PEG (in blue) and Au (in light red) were highlighted.

**Figure 5 nanomaterials-12-02053-f005:**
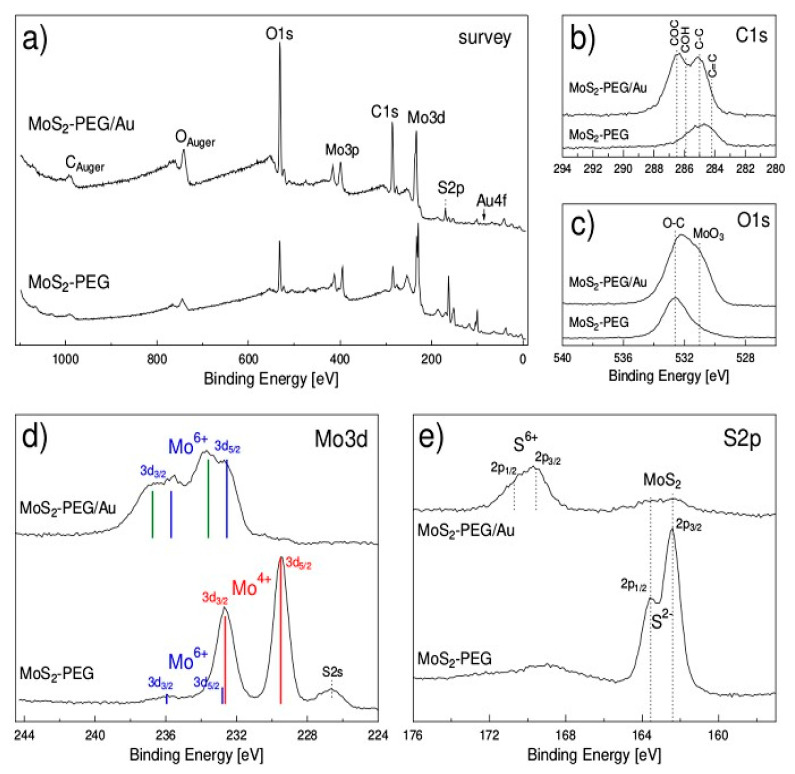
X-ray photoelectron spectra of MoS_2_-PEG and MoS_2_-PEG/Au samples: survey (**a**), C1s (**b**), O1s (**c**), Mo3d (**d**) and S2p (**e**).

**Figure 6 nanomaterials-12-02053-f006:**
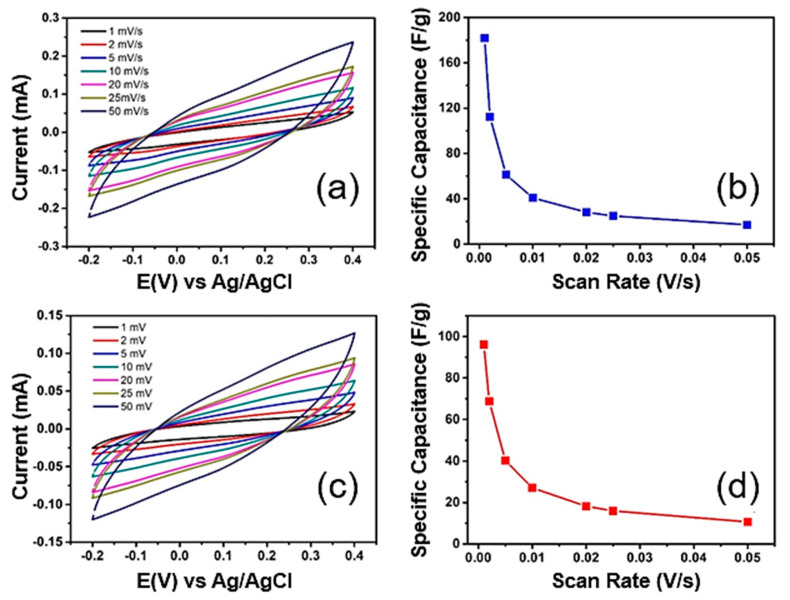
Cyclic voltammograms recorded with (**a**) MoS_2_-PEG and (**c**) MoS_2_-PEG/Au electrodes at different scan rates in 1.0 M Na_2_SO_4_ electrolyte solution and their respective profiles of specific capacitance (**b**,**d**).

**Figure 7 nanomaterials-12-02053-f007:**
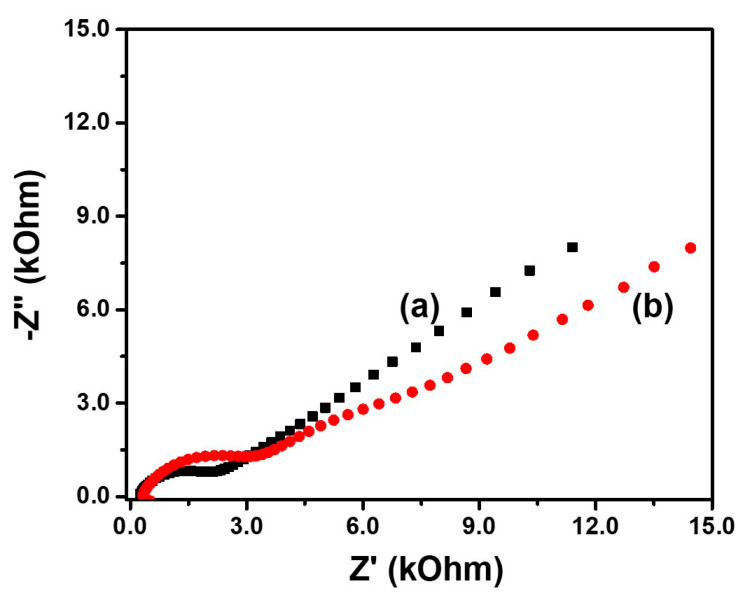
Nyquist plots of MoS_2_-PEG (**a**) and MoS_2_-PEG/Au (**b**) electrodes measured with [Fe(CN)_6_]^3−^/[Fe(CN)_6_]^4−^ 2 mM in KCl 0.1 M electrolyte.

**Figure 8 nanomaterials-12-02053-f008:**
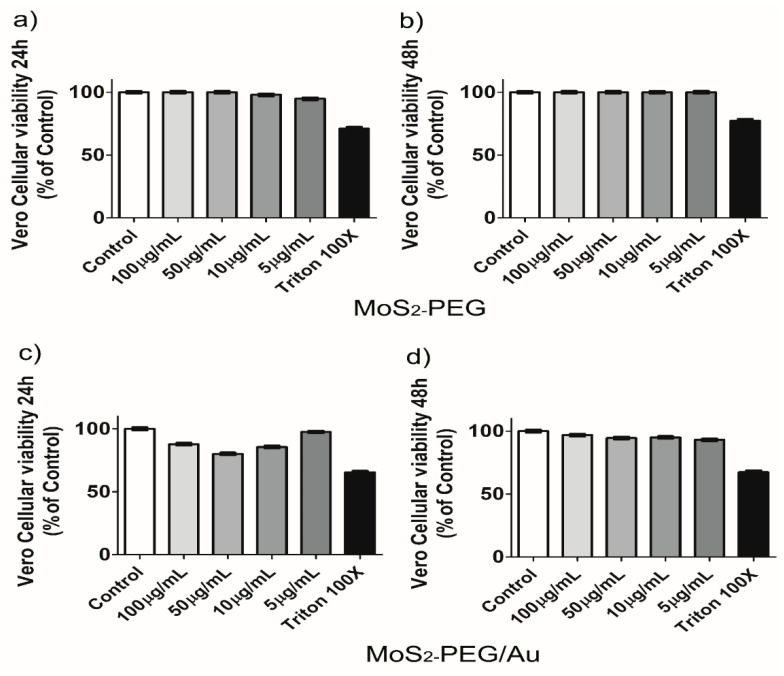
Vero cellular viability after 24 h and 48 h. Panels (**a**–**d**) show the percentage of Vero cellular viability exposed to different concentrations of MoS_2_-PEG and MoS_2_-PEG/Au. Results of 2 independent experiments in triplicate. (Mean ± SD).

**Figure 9 nanomaterials-12-02053-f009:**
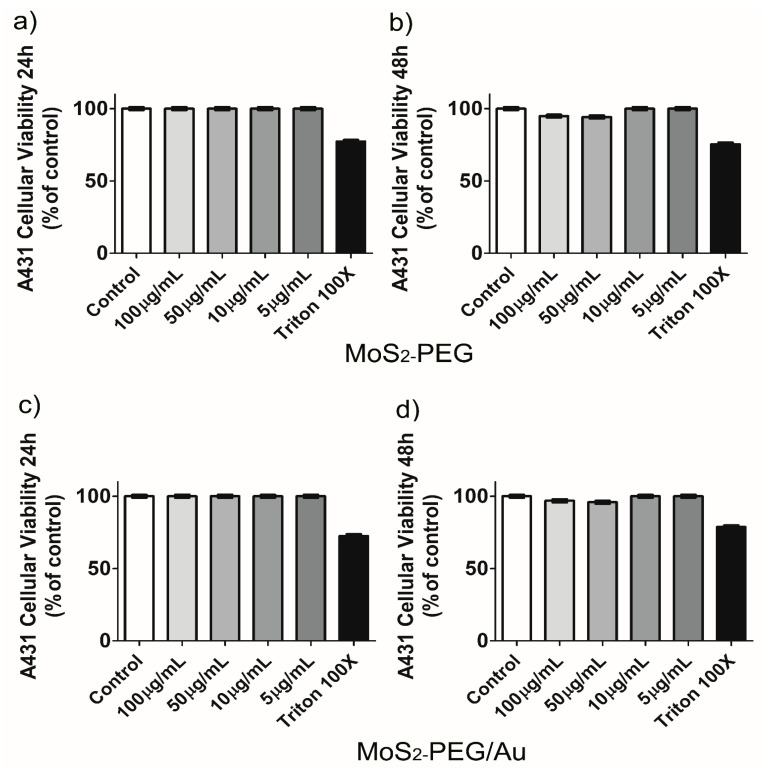
A431 cellular viability. Panels (**a**–**d**) show the percentage of cellular viability of these cells exposed to different concentrations of MoS_2_-PEG and MoS_2_-PEG/Au. Results representative of 2 independent experiments in triplicate. (Mean ± SD).

**Figure 10 nanomaterials-12-02053-f010:**
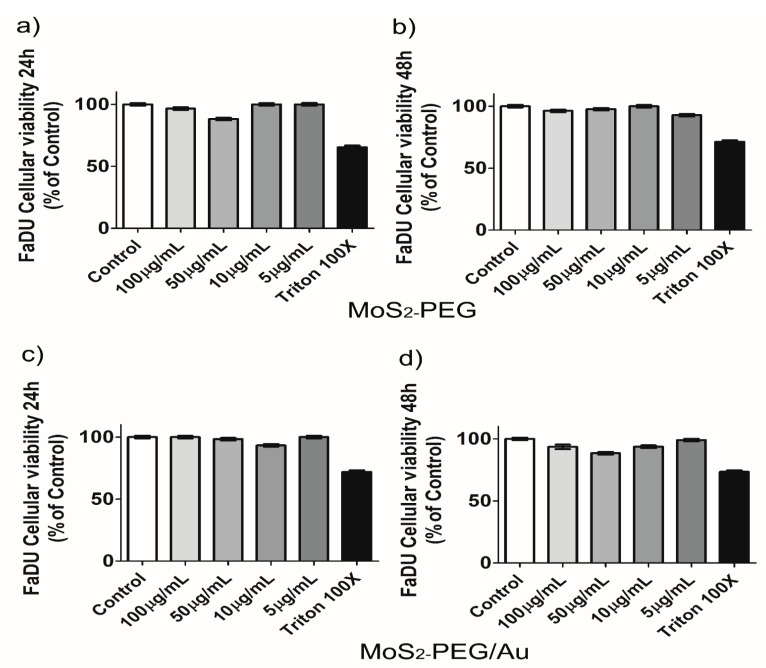
FaDU cellular viability. Panels (**a**–**d**) show the percentage of cellular viability of these cells exposed to different concentrations of MoS_2_-PEG and MoS_2_-PEG/Au. Results representative of 2 independent experiments in triplicate. (Mean ± SD).

**Figure 11 nanomaterials-12-02053-f011:**
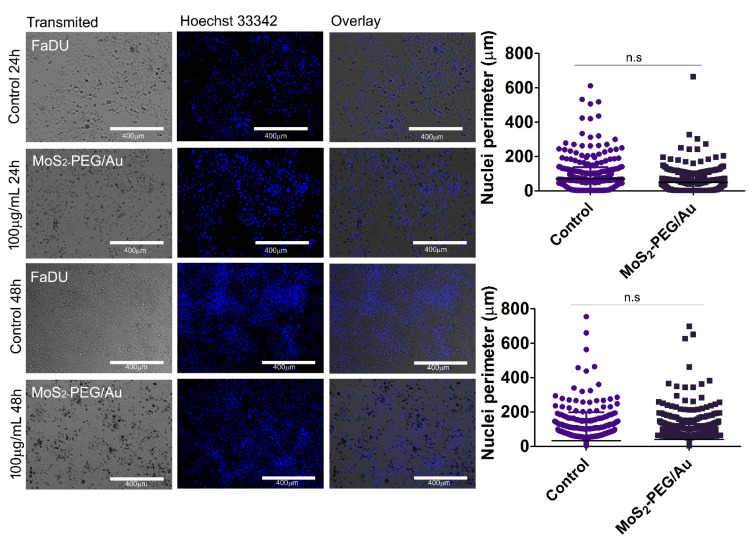
FaDU nuclei integrity. Representative fluorescent image of FaDU cells exposed to 100 µg/mL of MoS_2_-PEG/Au. Nuclei were stained with Hoechst 33,342. Data from 2 replicates (n.s = no significant). Bar scale = 400 µm and objective of 10×.

**Table 1 nanomaterials-12-02053-t001:** Electric properties of the MoS_2_-PEG and MoS_2_-PEG/Au nanostructures.

	Samples	
Electrical Parameters	MoS_2_-PEG	MoS_2_-PEG/Au
Sheet resistance (kΩ sq^−1^)	671.4	218.1
Resistivity (Ω m)	1342.7	436.2
Conductivity (10^−6^ S cm^−1^)	7.4	22.9

**Table 2 nanomaterials-12-02053-t002:** Values of contact angle of graphite sheet, MoS_2_-PEG and MoS_2_-PEG/Au nanosheets surfaces.

Materials	Contact Angle (°)
Graphite (substrate)	58.1 ± 3.9
MoS_2_-PEG	30.7 ± 3.4
MoS_2_-PEG/Au	37.1 ± 2.1

## Data Availability

Data presented in this article are available at request from the corresponding author.

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
