# Peer review of "Multifunctional Hybrid MoS2-PEGylated/Au Nanostructures with Potential Theranostic Applications in Biomedicine"

_nanomaterials, 2022, doi:10.3390/nano12122053_

Round 1

Reviewer 1 Report

In this work, the authors reported a kind of flower-like MoS2-PEG/Au microsphere for potential theranostic application. The MoS2 -PEG/Au microspheres were well characterized. However, this manuscript needs major revision before it can be published.

Detailed comments:

1.     As shown in SEM images, the MoS2-PEG/Au microspheres performed an aggregation state. Are nanoparticles in this state suitable for in vivo application?

2.     This work only verifies the biocompatibility of MoS2-PEG/Au microspheres. Are there any other advantages of the microspheres for biomedical applications, such as photothermal effect and so on?

3.     In Figures 8-10, SD values are missing in most groups. The cellular viability did not decrease with the increase of drug concentration, please explain this phenomenon.

There are some detail problems in this manuscript. For example, the font sizes in Figure 6 are inconsistent; Figure 11 is lack of labeling.

Author Response

Attached file

Reviewer 2 Report

The manuscript presents a comprehensive experimental analysis of molybdenum disulfide PEGylated/Au nanorods composite nanomaterials that might be used as novel theranostic material. The experimental data outlined in the manuscript is valuable for further studies of the composite materials and deserves to be published. However, I would recommend revision of the manuscript before it can be accepted for publication. Specifically, I would recommend the authors to address the following questions:

1.     The title and the abstract/introduction imply that these nanomaterials can be used for theranostic applications in biomedicine. However, there is no discussion about how this can be achieved. Would this composite nanomaterial be irradiated with IR laser? How the imaging with this material be possible? Is some kind of fluorescence be used or expected from these materials? What are the spectral frequencies for heating/ablation and/or imaging might be expected?

2.     Two composite materials were fabricated and characterized:  (a) molybdenum disulfide PEGylated/Au nanorods and molybdenum disulfide PEGylated. What is the difference and advantages/disadvantages between these two materials in terms of  theranostic applications in biomedicine?

3.     I would suggest including Fig. S1 in the supplementary materials in the manuscript (not as supplemental materials). Furthermore, I would suggest cutting the UV-vis spectrum in the UV region and presenting it within 400 nm – 1000nm spectral region. This way, the two SPR modes for gold nanorods would be more pronounced. Also, I would suggest providing the width of the nanorods and their aspect ratio as this would give some information about the two SPR peaks that are expected for nanorods.

Author Response

Attached file

Round 2

Reviewer 1 Report

This work can be acceptted now.